# The response of annual minimum temperature on the eastern central Tibetan Plateau to large volcanic eruptions over the period 1380–2014 AD

Yajun Wang[1,3], Xuemei Shao[1,2], Yong Zhang[1], Mingqi Li[1]

[1] Key Laboratory of Land Surface Pattern and Simulation, Institute of Geographic Sciences and Natural Resources, Chinese Academy of Sciences (CAS), Beijing, 100101, China
[2] University of Chinese Academy of Sciences, Beijing, 100049, China
[3] CAS Center for Excellence in Tibetan Plateau Earth Sciences and Institute of Tibetan Plateau Research, Chinese Academy of Sciences (CAS), Beijing, 100101, China

*Correspondence to*: Yong Zhang (zhangyong@igsnrr.ac.cn)

**Abstract.** Volcanic eruptions have a significant impact on global temperature; their consequences are of particular interest in regions that are especially sensitive to climate change, like the Tibetan Plateau. In this study, we develop a temperature-sensitive tree-ring-width standard chronology covering the period 1348–2014 AD using Qilian juniper (*Sabina przewalskii* Kom.) samples collected from the Animaqin Mountains on the Tibetan Plateau. We reconstruct the annual (prior August to current July) mean minimum temperature ($T_{min}$) since 1380 AD and show that our reconstruction explains 58% of the variance during the 1960-2014 calibration period. Our results demonstrate that in 77.8% of cases in which a volcanic eruption with a volcanic explosivity index of 5 or greater occurs, temperature decreases in the year of or the year following the eruption. The results of the Superposed Epoch Analysis also indicate that there is a high probability that the $T_{min}$ decreases for 2 years of a large volcanic eruption, especially when such eruptions occur in the Northern Hemisphere.

**Keywords.** Tibetan Plateau, tree rings, minimum temperature, volcanic eruption

## 1 Introduction

Large volcanic eruptions can affect the climate of the Earth (Robock, 2000) and have played a major role in past global temperature changes (Salzer and Hughes, 2007). Eruptions emit large amounts of ash particles and gases into the atmosphere, much of which are carried to other regions by atmospheric movement. These materials efficiently reflect incident solar radiation, resulting in the cooling of the Earth's surface. However, volcanic eruptions of similar magnitudes do not necessarily result in cooling across all areas of the world. For example, the 1991 eruption of Mount Pinatubo, Philippines, caused summer cooling over much of the globe in 1992, but the

temperature in some areas was above average (Robock and Mao, 1992). Thus, it is not necessarily clear to what extent or in what manner a strong volcanic eruption will influence temperature in a particular region.

Often referred to as the "third pole", the Tibetan Plateau (TP) is especially sensitive to climate change (Yao et al., 2012) and may therefore be more profoundly influenced by volcanic eruptions. As early as 1985, studies of the relationship between large volcanic eruptions (dust veil index > 1000 $m^3$) and temperature variations from 1951 to 1980 in China (Zhang and Zhang 1985) demonstrated that temperature on the northeastern TP decreased 8, 15, and 18 months after eruptions. However, temperatures in the 6 months immediately following an eruption were found to be relatively high. Jia and Shi (2001) studied climate signals following volcanic eruptions from 1950 to 1997 and found that temperatures on the TP dropped for 2 years following eruptions. However, new research focusing on regional differences over China found that winter volcanic eruptions occurring between 1956 and 2005 led to extensive warming of winter temperatures over the Tibetan Plateau (Sun et al., 2019). These studies show that temperature on the TP is affected by volcanic activity, but it is important to note that they are based on instrumental data, which covers a relatively short time span. Temperature changes caused by strong volcanic eruptions can affect tree growth (Tognetti et al., 2012). This influence that can be seen in the rings of certain trees (D'Arrigo et al., 2013; Filion et al., 1986; Lamarche and Hirschboeck, 1984) and used to identify past volcanic activity (D'Arrigo et al., 2013; Filion et al., 1986; Lamarche and Hirschboeck, 1984). Especially when long-lived trees are available, tree rings can serve as temperature-sensitive proxies for investigating climate responses to volcanic eruptions that occurred prior to the instrumental record (D'Arrigo et al., 2013; Salzer and Hughes, 2007).

Tree rings from the TP can potentially be used to study the regional response of climate to volcanic activities. Previous tree-ring-based studies have shown that some cold years are closely correlated with large volcanic events (Liang et al., 2008; Liang et al., 2016; Zhang et al., 2014). Li et al (2017a) quantitatively assessed the correlation between temperature changes on the southeastern TP and volcanic eruptions and showed that most of the years of extreme cold in the past 304 years occurred 1–2 years after major volcanic eruptions. However, the influence of volcanic eruptions on temperatures on the TP over the long-term is less well understood due to the paucity of data in this region.

The Animaqin Mountains are located on the northeastern TP and have yielded many long tree-ring series (Chen et al., 2016; Gou et al., 2007; Gou et al., 2008; Gou et al., 2010). These series can provide a significant record of how tree growth responds to volcanic eruptions, but such studies are still rare across this area of the TP. Using tree-ring samples of Qilian juniper (*Sabina przewalskii* Kom.) collected from a new sample site in the southeastern part of the Animaqin Mountains, we develop a 667-year tree-ring-width chronology. We then use the chronology to

reconstruct annual mean minimum temperatures ($T_{min}$) across the east-central TP. Finally, we explore the response of $T_{min}$ to strong volcanic eruptions (Volcanic Explosivity Index (VEI) > 4) over the past six centuries.

## 2 Data and methods

### 2.1 Tree-ring data

Qilian juniper samples were taken from a natural woodland 35 km west of Ningmute town, Henan County,
Qinghai Province, China (E100.96, N34.62, 3806 m a.s.l). This area is located in the sub-frigid zone and has a semi-humid climate (Zheng et al., 2013). According to climate data of the Henan meteorological station (1960–2015; Table 1), mean annual temperature is 0.31℃ and mean annual precipitation is 582 mm. Precipitation is mainly concentrated between May and September. Because the site is close to the Yellow River and the soil layer in the forest area is thick, the moisture conditions of the forest are good. The regional vegetation zone is coniferous
forest and the main tree species include *Sabina przewalskii Kom.*, *Picea* crassifolia Kom., *Betula spp.*, and *Salix cheilophila*. The study area is located on the southern slope of the Animaqin Mountains and ranges in elevation from 3523 to 3900 m a.s.l. (Figure 1). The gradient of the slope is 30°–40°. A total of 110 cores from 55 trees were drilled at breast height with an increment borer in 2015.

(figure 1 near here)

The cores were air-dried, fixed to wooden mounts, polished, and cross-dated. The cores were then measured using a Lintab 6 tree-ring-width measuring instrument with a resolution of 0.01 mm. The COFECHA program (Holmes, 1983) was used to check the quality of the cross-dating and the accuracy of the measurements. The signal-free standardization method (Melvin and Briffa, 2008) was adopted to standardize the tree-ring data in order to minimize trend distortion in the chronologies produced by the straight-line detrending method. The RCSigFree
program (https://www.ldeo.columbia.edu/tree-ring-laboratory/resources/software) was used to establish the tree-ring-width chronology. In this procedure, the age trend curve was fitted to a cubic smoothing spline with a 50% cutoff at approximately 67% of the mean segment length. The validity of the chronology was assessed using the mean correlation coefficient for the tree-ring series (Rbar) and the value of the expressed population signal (EPS).

### 2.2 Instrumental data

Thirteen meteorological stations were identified in the region of the sampling site: Zhongxinzhan (ZXZ), Dari (DR), Xinghai (XH), Guoluo (GL), Tongde (TD), Guinan (GN), Zeku (ZK), Henan (HN), Jiuzhi (JZ), Tongren (TR), Maqu (MQ), Langmusi (LMS), and Hezuo (HZ). For each station, data for four climate parameters – monthly

total precipitation (P), monthly mean temperature ($T_{mean}$), monthly mean minimum temperature ($T_{min}$), and monthly mean maximum temperature ($T_{max}$) – were extracted from the China Meteorological Data Sharing Service System (http://www.cma.gov.cn/2011qxfw/2011qsjgx). Details are provided in Table 1.

(Table 1 near here)

Because the observation intervals of the stations shown in Table 1 differ, it was necessary to select the longest record to ensure the stability of the correlation function. Seven stations with instrumental data spanning <30 years were excluded: GL, GN, and TR (all of which were established in the 1990s), and ZXZ, TD, ZK, and LMS (all of which ceased monitoring in the 1980s or 90s). Climate data from the other six stations (DR, XH, HN, JZ, MQ, and HZ) were used for the following analysis (Fig. 1). During the quality control of the data from the six stations, we found that there are some problems in the instrumental data from the HN, JZ, and XH stations. For example, there are some missing values in April and May in 1962 at JZ station. To resolve this problem, we used the interpolation approach to estimate the missing values based on the complete instrumental records from nearby stations (see the supplementary section for details). Finally, taking the means for the interval 1971–2000 as references, we calculated the precipitation departure percentages and average temperature departures for all six stations. The mean values are considered to be representative of the regional climate. The monthly climate data from previous July to current September were used to analyze the response of tree growth to climate change.

### 2.3 Volcanic data

Data about volcanic activity were obtained from the Volcanic Explosivity Index (VEI) sequence published by the Smithsonian Institution Global Volcanism Program (http://volcano.si.edu). Globally, there have been 46 strong volcanic eruptions (VEIs> 4) since 1380 AD. For this study, it is important to note that eruptions that occurred in October, November, or December of a given year are marked as being a volcanic event in the following year. This is because tree radial growth mostly stops in October on the TP. We developed 7 sets of volcanic data for further analysis based on geography: (a) global; (b) Northern Hemisphere; (c) Southern Hemisphere; (d) low latitude 30°S–30°N; (e) Northern Hemisphere mid-latitude 30°N–60°N; (f) Southern Hemisphere mid-latitude 30°S–60°S; and (g) Northern Hemisphere high latitude.

### 2.4 Methods

The correlation function was used to analyze the relationship between the tree-ring-width index and climatic factors. A fitted equation was then established using a simple linear regression and verified by the cross-validation method (Michaelsen, 1987) and by split-period calibration/verification analysis (Meko and Graybill, 1995). Several

statistical tests (sign test, product mean value, reduction error, and coefficient of efficiency) were utilized. The split periods for calibration were 1960–1991 and 1984–2014. The correlations between the observed and reconstructed series and the gridded dataset (TS3.22; Mitchell and Jones, 2005) from the University of East Anglia Climatic Research Unit (CRU) were analyzed using the KNMI Climate Explorer research tool (http://climexp.knmi.nl). The Superposed Epoch Analysis (SEA) method (Haurwitz and Brier, 1981) was used to analyze the influence of volcanic activity on regional temperature. SEA is a statistical method used to resolve significant signal-to-noise problems and is often used to study the link between climate and discrete events such as solar activity, fire events, volcanic activity, etc. (Adams et al., 2003; Singh and Badruddin, 2006; Swetnam 1993; Esper et al., 2013;). In this study, the year of a volcanic eruption is regarded as year 0. The years before the volcanic eruption are denoted as -1, -2, -3 and so forth, whereas the years after the eruption are expressed as 1, 2, 3, etc. The impact of volcanic eruptions on temperature on the east-central TP was analyzed by comparing differences in temperature in the years leading up to and following an eruption. The significance of responses was determined by a Monte Carlo resampling procedure (10,000 iterations) (Adams et al., 2003).

## 3 Results

### 3.1 Statistical characteristics of the chronology

The tree-ring-width chronology covers the time period from 1348 to 2014 AD (Figure 2a) and has an average length of 325.1 years. The average sensitivity is 0.171; the first-order autocorrelation coefficient is 0.261. The signal-to-noise ratio is 21.870, the EPS is 0.956, and the Rbar ranges between 0.31 and 0.489. The chronology is considered reliable (EPS > 0.85) from 1380 AD, at which point the sample depth is nine.

(Figure 2 near here)

### 3.2 Correlations between the tree-ring-width index and climate factors

Figure 3a shows that the tree-ring-width index correlates positively and significantly with precipitation in February, and has a positive correlation with $T_{mean}$, $T_{max}$, and $T_{min}$, most notably with $T_{mean}$ and $T_{min}$. For the 15-month period from previous July to current September, the tree-ring-width index correlates significantly and positively with the monthly $T_{min}$ except during previous August. Correlation with annual $T_{min}$ (from previous August through current July; hereafter referred to as $T_{min}87$) is at the 0.01 significance level.

The first-differenced correlations between the tree-ring-width index and climate data are weak; in some months, the first-order correlations are even negative (Figure 3b). The first-differenced chronology correlates significantly

and positively with precipitation of the previous September and the current May, but negatively with precipitation of the previous December. For temperature, the tree-ring-width index shows clear correlations with $T_{mean}$ and $T_{min}$ and correlates significantly and positively with $T_{min}$ of previous September and November and current February, June, and July. The positive correlation with the annual $T_{min}$ is at the 0.01 significance level.

The tree-ring width index is most strongly correlated with $T_{min}87$ (r = 0.767, p<0.001). This correlation remains strong and positive after first-differencing (r = 0.583, p<0.001), which indicates that the tree-ring-width index is suitable for reconstructing the $T_{min}87$ for the given period.

(Figure 3 near here)

**3.3 Reconstruction development and verification**

The tree-ring-width index for the current year ($SF_t$) was selected as the predictor to reconstruct the annual mean minimum temperature departure ($T_{min}87$):

$$T_{min}87 = -2.497 + 2.348 SF_t. \tag{1}$$

The reconstruction accounts for 58% of the variance in the instrumental series during the calibration period (1960–2014). The $F$ value is 74.187, and exceeds the 0.001 confidence level. The transfer function is therefore highly significant (Figure 4a and 4b).

(Figure 4 and Table 2 near here)

Cross-validation for the calibration period 1960–2014 shows that the sign test (ST) and the first-difference sign test (FST) are all significant at the 0.01 level (Table 2). ST/FST are also significant for the split periods 1960–1991 and 1984–2014, indicating that the reconstructed $T_{min}87$ is in good agreement with the instrumental values for the split periods and for the overall period. The product mean value of 5.13 (p < 0.01) implies favorable coincidence in the two series. A reduction error (RE) value of 0.557 for the whole 1960–2014 period indicates a high degree of similarity between the reconstructed and instrumental series. Thus, Equation (1) is suitable for reconstructing changes in $T_{min}87$.

**3.4 $T_{min}87$ change since 1380 AD**

The reconstructed $T_{min}87$ from 1380–2014 AD (Figure 4c) shows clear interannual variations in temperature. The mean of the reconstructed $T_{min}87$ is –0.15°C (standard deviation $\sigma = 0.41$). $T_{min}87$ values below the mean 1.5$\sigma$ are defined here as 'extreme cold', and values above the mean 1.5$\sigma$ as 'extreme warm'. There are 39 extreme cold years; the five coldest are 1488, 1490, 1824, 1862, and 1872. There are 47 extreme warm years; the five warmest

are 1418, 1996, 1999, 2009, and 2010. Cold periods lasting longer than 5 years are 1483–1495, 1555–1568, 1586–1602, 1686–1696, 1840–1854, 1872–1876, 1893–1901, 1910–1920, 1961–1968, and 1975–1983. Of these, 1586–1602 was the longest period and experienced the lowest temperatures, followed by 1840–1854. Warm periods lasting more than 5 years were 1409–1424, 1504–1509, 1655–1659, 1729–1739, 1775–1779, 1781–1813, and 1991–2009. The longest warm period was 1781–1813 and the warmest interval was 1991–2009, followed by 1409–1424. Changes at both high and low frequencies over the study period clearly indicate that the climate on the TP has been warming, especially since the 1980s.

### 3.5 Spatial representation

Correlations between the reconstructed series, instrumental data, and the CRU gridded annual $T_{min}$ are high over most of China and even across most of Asia for the period 1960–2014 (Figure 5a and 5b). However, the first-differenced instrumental data only correlate significantly with the CRU gridded data over the TP (Figure 5c). The coverage of the significant first-differenced correlations of reconstructed $T_{min}$87 with the CRU grid data is smaller, and is mainly located on the central and western part of the TP, especially around the Animaqin Mountains (Figure 5d).

(Figure 5 near here)

### 3.6 Relationship between volcanic activity and minimum temperature

A comparison of large volcanic eruption events and the reconstructed $T_{min}$87 is shown in Figure 6. For most years, large volcanic eruptions (VEIs> 4) coincide with a drop in $T_{min}$ across the study area. Of the 46 strong volcanic eruption years, there are 35 event years in which temperature decreases in the year of the eruption or 1, 2, or even 3 years after the event.

(Figure 6 and Figure 7 near here)

Over the period 1380–2014 AD, $T_{min}$ on the TP has been greatly influenced by large volcanic eruptions, especially those occurring in the Northern Hemisphere. Globally, $T_{min}$ decreases in the year of an eruption and for one to two years thereafter (Figure 7a). In the year immediately following an eruption, the relationship is at the 0.05 significance level. Eruptions in the Northern and Southern Hemispheres (Figure 7b and 7c) also coincide with a $T_{min}$ decrease in the year of the eruption and for one to two years thereafter. However, eruptions in the Northern Hemisphere have a more obvious influence on $T_{min}$ in the study area, as indicated by the strong drop in temperature

in the year following an eruption (Figure 7b). Southern Hemisphere mid-latitude eruptions occur at greater distances from the TP and therefore have a weaker influence on $T_{min}$.

$T_{min}$ in the study area decreases significantly in the years following volcanic activity at low latitudes (Figure 7d). Eruptions in the mid-latitudes of the Northern Hemisphere clearly coincide with drops in temperature, but the decreases are not statistically significant (Figure 7e). Eruptions in the mid-latitudes of the Southern Hemisphere and in the high latitudes of the Northern Hemisphere also coincide with reduced $T_{min}$ in the year of the eruption and in the following year, but the decreases are not statistically significant (Figure 7f and 7g).

## 4 Discussion

### 4.1 Reliability of the $T_{min}$ reconstruction

The correlation between the tree-ring width index and climate factors shows that the relationship between tree radial growth and precipitation is not statistically significant except in February. With the thick topsoil and humid climate in spring and summer, the study area meets the moisture needs of trees for radial growth. However, according to instrumental data from the GL weather station (Table 1), the elevation of which is close to that of the sampling site, the annual $T_{mean}$ and $T_{min}$ are 2.25°C and -6.76°C, respectively. These temperatures are quite low for tree growth, and the statistically significant positive correlation between growth and temperature shows that tree radial growth in this area is restricted by temperature, especially $T_{min}$. $T_{min}$ before and during the growing season may affect tree growth in several ways. In winter and early spring, warmer $T_{min}$ protects roots from cold damage and triggers earlier snowmelt. Warmer $T_{min}$ may therefore result in a longer growing season, and trees may experience increased growth during the subsequent growing season (Pederson et al., 2004; Fu et al. 2012; Hollesen et al. 2015; Williams et al. 2015). On the other hand, $T_{min}$ is known to positively affect conifer tracheid division and enlargement by controlling the onset and conclusion of xylogenesis during the growing season (Deslauriers et al., 2003; Rossi et al., 2008; Li et al., 2017b; Hosoo et al. 2002; Steppe et al., 2015).

It should be mentioned that the CE value of the validation period remains negative for the period from 1960 to 1983 even though we tested different meteorological stations and different observation intervals in the analysis process. One reason for this could be a distortion of the meteorological data due to the poor management and/or the relocation of meteorological stations during the 1950s and 1960s on the TP. However, the cross-validation results-indicate that the equation is otherwise reliable. The negative CE value should not influence our analysis of the high frequency climate response to volcanism.

The correlations of the reconstructed series with CRU $T_{min}$ reflect the regional significance of the reconstruction in general. However, the consistent warming trend in $T_{min}$ over most of Asia (Dong et al., 2017) may result in a large area of significant positive correlations. For this reason, the results of our first-order correlation analysis are therefore more dependable, i.e., our reconstruction reliably reflects temperature variations in the Animaqin Mountains and in the area to the west.

The reconstructed Tmin87 was further compared with minimum temperature reconstructions from other regions of the TP, including HY (northeastern TP; Zhang et al., 2014), QML(ZHD) (central TP; He et al., 2014), HBL (eastern TP; Gou et al., 2007), YS (south-central TP; Liang et al., 2008), and LIT(TAN) (southeastern TP; Li and Li, 2017). As a result of differences in reconstruction factors, study regions, and methods of chronology establishment, there are some differences between the chronologies. For example, there are significant regional

differences in $T_{min}$ reconstructions over the past 100 years. However, there is also notable consistency at the interannual-multidecadal scale. The correlation coefficients between our reconstruction and each sequence are 0.227 (HY; $p < 0.01$), 0.328 (QML(ZHD); $p < 0.01$), 0.499 (HBL; $p < 0.000$), 0.235 (YS; $p < 0.01$), and 0.317 (LIT(TAN); $p < 0.01$). The closer the sequence is to the study site, the more similar the changes in $T_{min}$ are. For example, the Tmin87 of this study is most strongly correlated with the HBL reconstruction (Figure 8d), which is

nearest to our study area. Both reconstructions show low-temperature periods at the end of the 16$^{th}$ century and from the 1670s to the 1720s. Although it is located further from the study area, the HY reconstruction also shows cold periods in the late 15$^{th}$ century, at the end of the 16$^{th}$ century, at the beginning of the 18$^{th}$ century, and in the middle of the 19$^{th}$ century (Figure 8a). Similarly, the low-temperature periods in the late 15$^{th}$ and late 16$^{th}$ centuries, the 1670s to 1720s, and the 1960s to 1980s are in agreement with those in the QML(ZHD) reconstruction (Figure

8b). The low-temperature interval in the 1960s–80s coincides with the cold interval in the YS reconstruction (Figure 8e). Finally, although the LIT(TAN) reconstruction is located in a more distant region of the TP, its low-temperature intervals are consistent with those in our reconstruction (e.g., the end of the 15$^{th}$ century, 1670s–1720s, 1900s–1920s, and 1960s–80s) (Figure 8f). It is interesting that higher Rbar values appear in some special intervals, i.e., the cold intervals of the 1460s-1500s and 1800s-1820s and the warmest period of 1980-2014. These higher Rbar

values indicate a good consistency among tree-ring series during these periods; in fact, these three intervals are evident in tree-ring-based studies from elsewhere on the Tibetan Plateau (Huang et al. 2019; Liang et al. 2016; Shi et al. 2019). The two cold periods identified in our series correspond to periods of weaker solar activity (the Spörer Minimum and the Dalton Minimum), and to a few very strong tropical eruptions (e.g., the Tambora eruption in 1815 and another stratospheric eruption in 1809) (Cole-Dai et al., 2009). Similarly, the warming in 1980-2014 is

closely related to the influence of human activities. These responses are indicative of the consistent response of tree growth to strong external forcing factors and of the reliability of our reconstruction.

### 4.2 The effect of volcanic activity on the reconstructed $T_{min}$87

As shown in Figure 6, the cooling probability in the year of or the year following a large volcanic eruption is very high. The effects of the Tambora volcano in 1815 (VEI = 7) were recorded in many parts of the world. Our reconstruction indicates that temperatures on the TP dropped by about 0.5°C in 1816, the year following the eruption. On the southeastern TP, the cold period from 1816 to 1822 may have also been related to the Tambora eruption (Liang et al., 2008). Other research in the northeastern TP has shown that cold years can be matched with known tropical volcanic events in 15 of 21 cases (Zhang et al., 2014). We compared the years of cooling we identified in this study with those identified by Zhang et al. (2014) and found that the cooling years are either the same or within a year of each other. Liang et al. (2016) showed that on the southeastern TP, the 15 coldest years of the past 304 years occurred mostly within 1-2 years of a major volcanic eruption; nine of those years are also seen in our study. The results of SEA analysis further confirm that the temperature on the TP is affected by strong volcanic eruptions, especially those occurring in the Northern Hemisphere and at low latitudes, and that cooling occurs in the study area within a year or two of a major eruption.

Studies have shown that some cold intervals on the eastern and southern TP may be influenced by large volcanic eruptions in low-latitude regions (Bi et al., 2020; Duan et al., 2019a; Krusic et al., 2015). Oman et al. (2005) used an ensemble simulation of the climate response to high-latitude volcanic eruption to show the surface air temperature in the TP were cooling in the first winter. Using the fully coupled NCAR Community Climate System Model (CCSM3), Schneider et al. (2009) found that there is significant summer cooling over the continents in the case of both tropical and high-latitude volcanism for the period 1250–1300 AD. New research by Toohey et al. (2019) demonstrated that explosive extratropical eruptions since 750 AD have produced stronger hemispheric cooling than tropical eruptions. These results are in alignment with ours, which show that strong eruptions in both the Northern Hemisphere and the tropics can lead to decreases in $T_{min}$ in our study area. Our results suggest that the effects of volcanic eruptions in the Northern Hemisphere are more pronounced. It should be mentioned that a one-to-one correspondence between large eruptions and cooling is not observed, temperature variability is also driven by factors such as circulation patterns like ENSO (Breitenmoser et al., 2012; D'Arrigo et al., 2011; Duan et al., 2019b). Cooling on the TP as a result of large volcanic eruptions may be weakened or masked by other influencing factors (Duan et al., 2018). Establishing a deeper understanding of the relationship between eruptions

and cooling will depend on expanding the spatial network of long-term, temperature-sensitive chronologies from the TP.

## 5 Conclusions

This study establishes a 667-year-long tree-ring-width chronology for the east-central TP. The anomaly sequence of the annual $T_{min}$ ($T_{min}87$) was reconstructed for the period 1380–2014 AD. The lowest temperatures with the longest duration occurred in 1586–1602; the longest and warmest interval was 1991–2009. The $T_{min}$ on the TP has been increasing, most notably since the 1980s. The SEA analysis shows that $T_{min}$ decreases in the year following strong volcanic activity in the Northern Hemisphere. Thus, the climate across the east-central TP is sensitive to large-scale background climatic factors such as volcanic activity. Further investigations are needed to fully understand the connection between volcanic eruptions and temperature on the TP, and to incorporate this information into regional predictions of annual and decadal temperatures.

## Author contributions

Yong Zhang and Xuemei Shao designed this study and took the tree-ring samples; Yajun Wang and Yong Zhang analysed the data and prepared the manuscript. Mingqi Li prepared the volcanic data and took the tree-ring samples.

## Acknowledgements

We thank two anonymous reviewers and the guest editor Kevin Anchukaitis for their helpful comments. This study was supported by the National Key R&D Program of China [Grant No. 2016YFA0600401] and the National Natural Science Foundation of China (Grant No. 41630529 and 41430528).

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

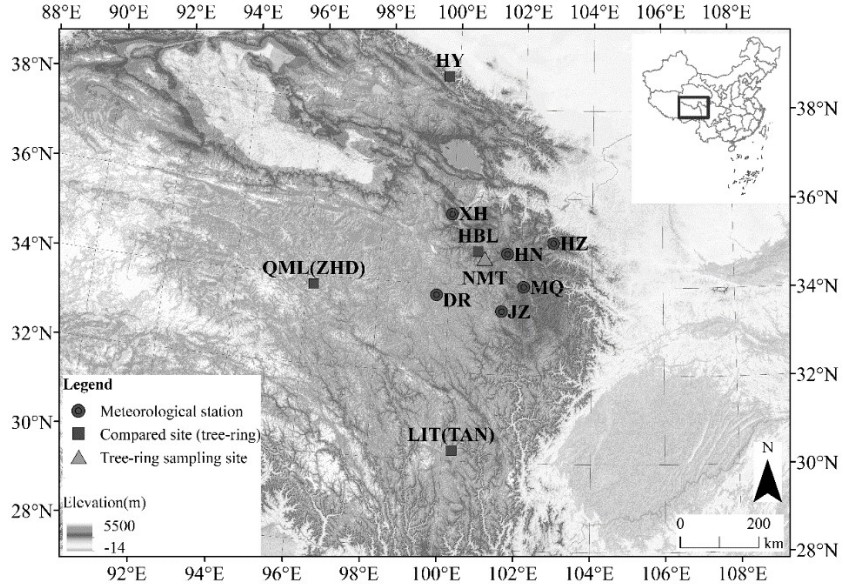


Figure 1. Locations of the sampling sites, meteorological stations, and compared sites. The digital elevation model is provided by Geospatial Data Cloud site, Computer Network Information Center, Chinese Academy of Sciences (http://www.gscloud.cn).


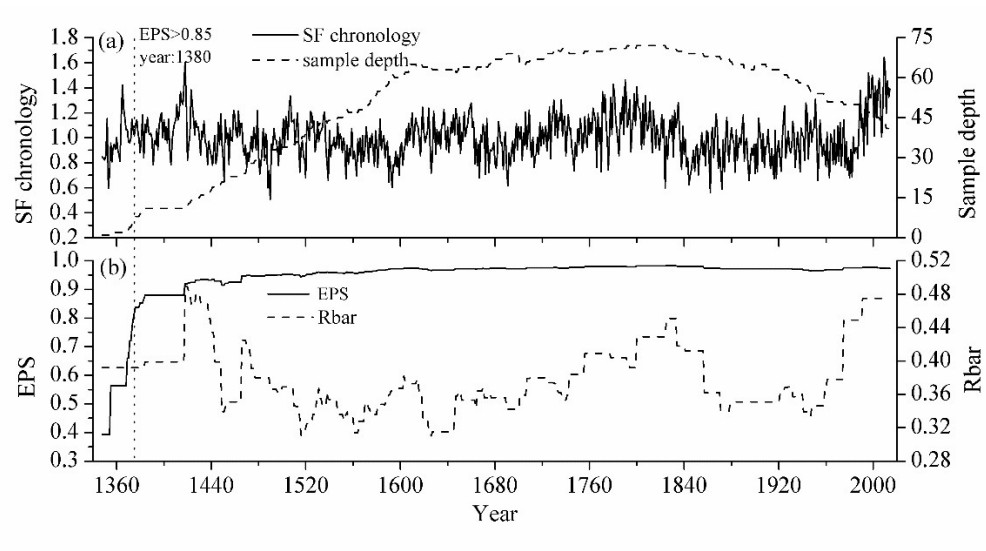

Figure 2. (a) Tree-ring-width chronology; (b) EPS and Rbar values. The dotted vertical line denotes the year 1380 CE, which is when the EPS value exceeds the 0.85 threshold.

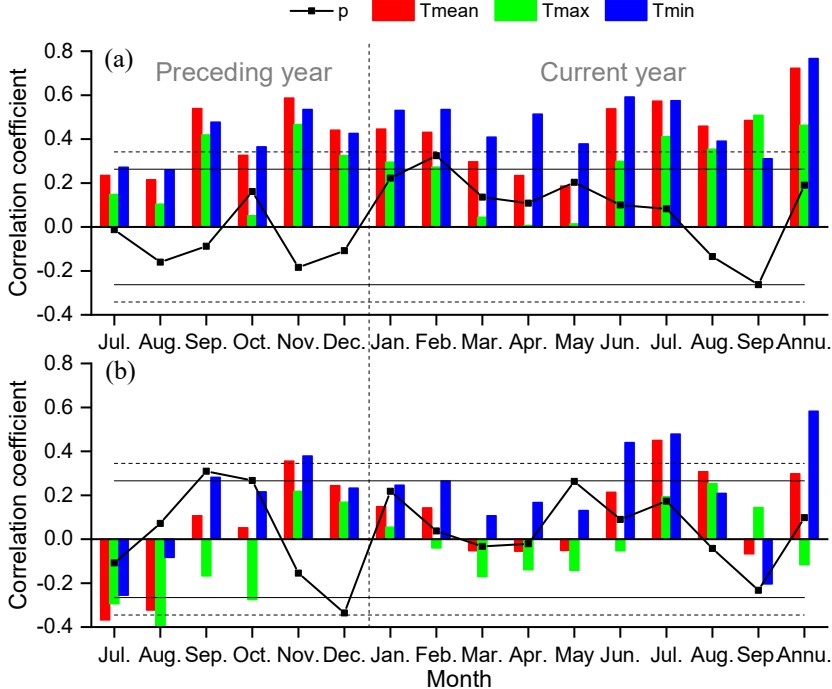

Figure 3. Correlations between (a) the tree-ring width index and climate data and (b) the first-differenced tree-ring-width index and climate data. The horizontal solid lines indicate the 0.05 significance level; the horizontal dashed lines show the 0.01 significance level. Annu. represents the entire period from previous August to current July.

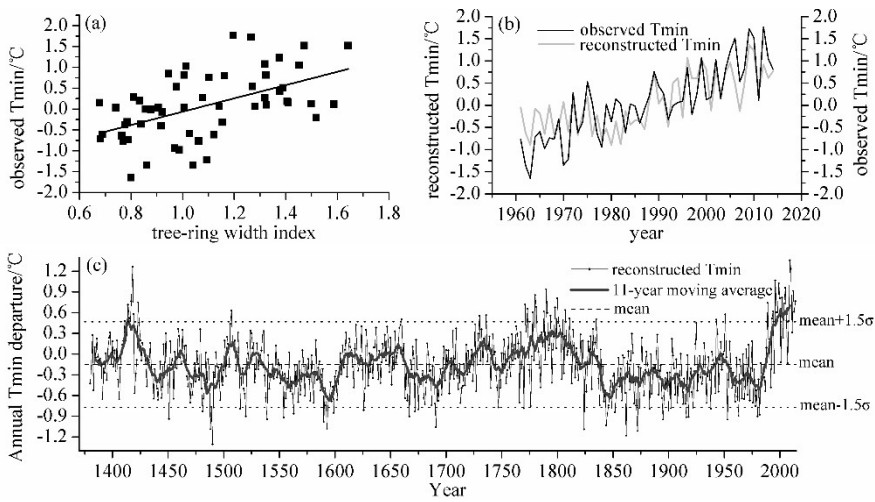

Figure 4. (a) Scatter plot of the tree-ring index and the instrumental annual $T_{min}$. (b) Instrumental (black line) and reconstructed (gray line) annual $T_{min}$. (c) Changes in annual $T_{min}$ since 1380 AD and the 11-year moving average (black solid line).

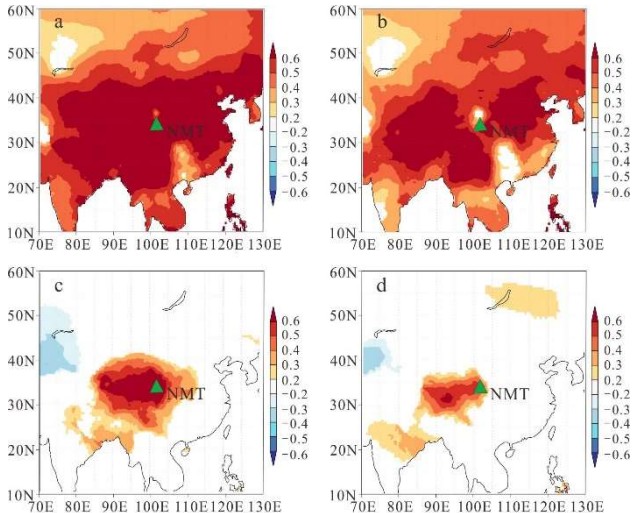

**Figure 5. Correlations of CRU $T_{min}87$ with $T_{min}87$: (a) instrumental; (b) reconstructed; (c) first-differenced instrumental; and (d) first-differenced reconstructed.**

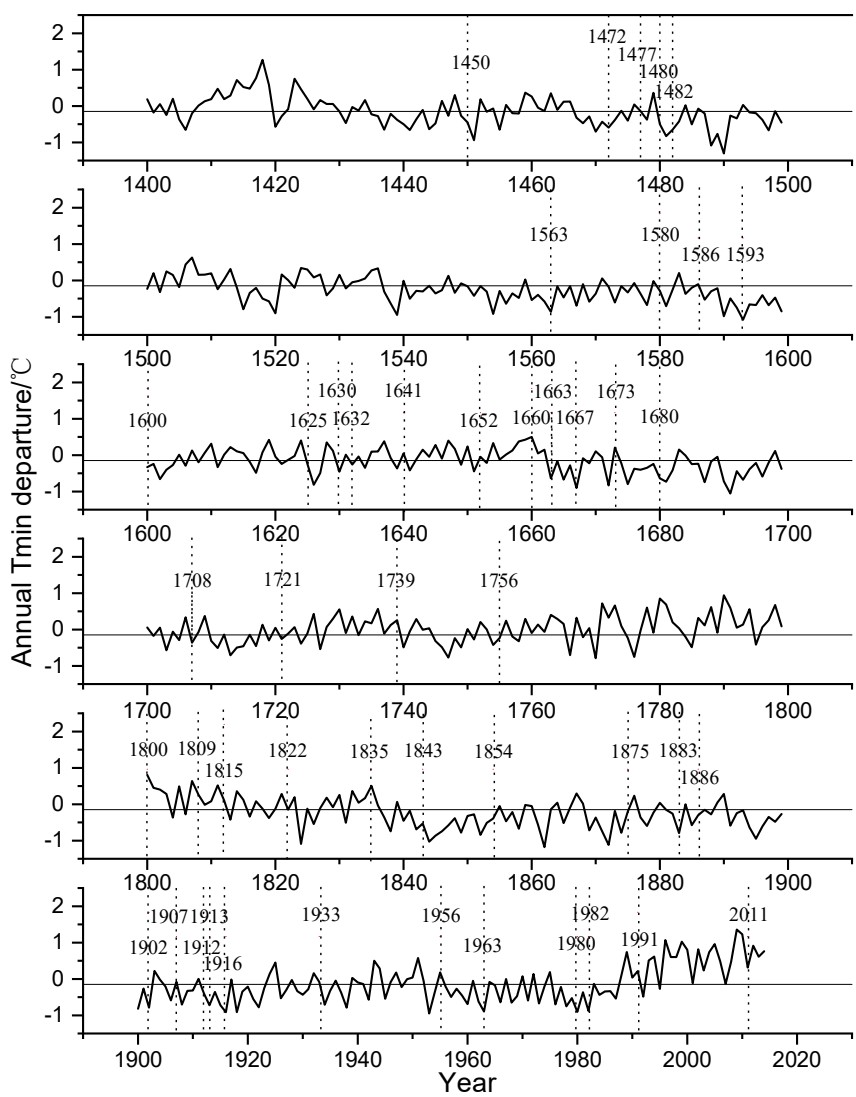

**Figure 6.  Plot of Tmin reconstruction since 1400 CE. The x-axis in each sub-figure indicates the year from the 15th century (top figure) to the 20th century (bottom figure). The dotted lines and the numbers indicate the years in which the 46 volcanic events used in this study occurred.**

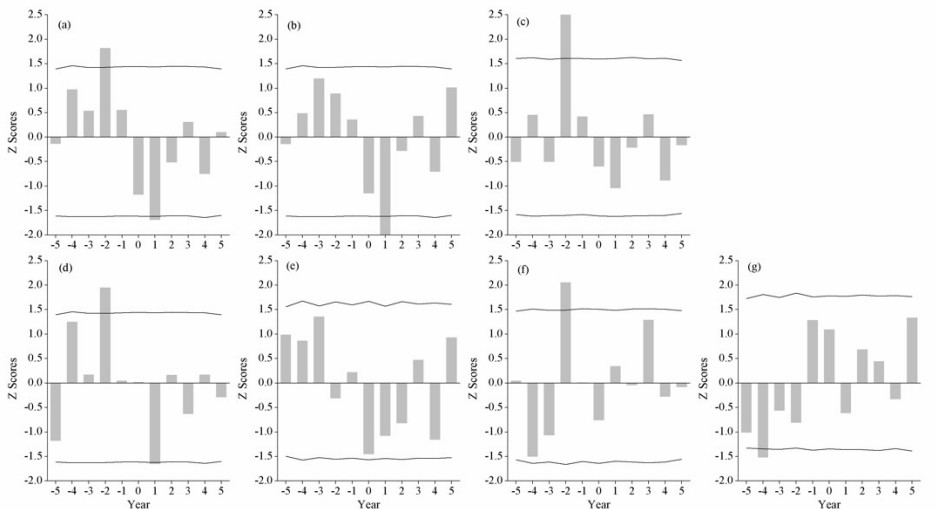

**Figure 7. SEA results of our reconstructed $T_{min}87$ with large volcanic eruptions in different areas: (a) global; (b) Northern Hemisphere; (c) Southern Hemisphere; (d) low latitudes 30°S–30°N; (e) Northern Hemisphere mid-latitude 30°N–60°N; (f) Southern Hemisphere mid-latitude 30°S–60°S; and (g) Northern Hemisphere high latitude. 0 = year of volcanic eruption; −1 = year before eruption; 1 = year after eruption. The solid lines represent the 95% confidence limits using Monte Carlo type block bootstrapping (Adams et al., 2003).**

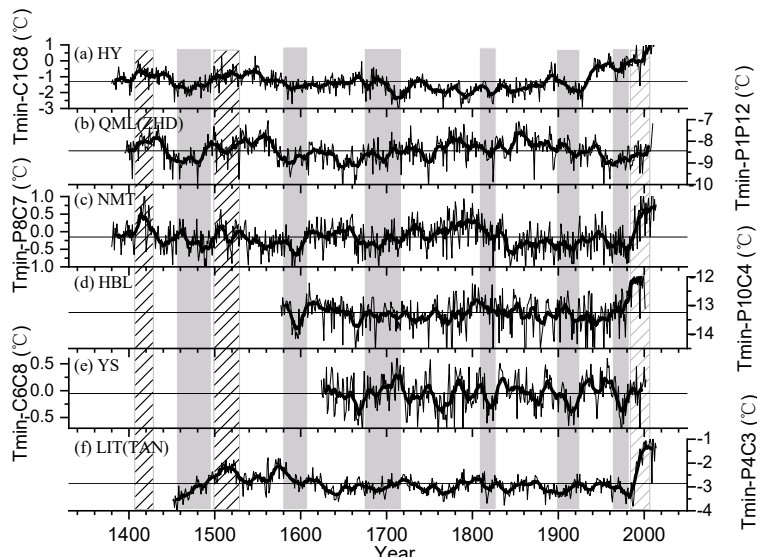

**Figure 8. Comparison of our reconstructed $T_{min}87$ with other $T_{min}$ reconstructions from the TP for the period 1380–2014 AD. (a) HY, January-August $T_{min}$ on the northeastern TP (Zhang et al., 2014); (b) QML(ZHD), January–December $T_{min}$ on the central TP (He et al., 2014); (c) Our study; (d) HBL, October–April $T_{min}$ on the eastern TP (Gou et al., 2007); (e) YS, June–August $T_{min}$ on the south-central TP (Liang et al., 2008); (f) LIT(TAN), April-March $T_{min}$ on the southeastern TP (Li and Li, 2017). Columns with slash marks indicate warm periods in our reconstruction; solid grey columns indicate cold periods.**

**Table 1. Meteorological station details**

| Station | Longitude | Latitude | Elevation (m) | Observation time span | Annual P (mm) | Annual $T_{mean}$ (°C) | Annual $T_{max}$ (°C) | Annual $T_{min}$ (°C) |
|---------|-----------|----------|---------------|----------------------|---------------|-----------------------|----------------------|----------------------|
| ZXZ | 99°12′ | 34°16′ | 4211.1 | 1959–1997 | 459.1 | −3.85 | 3.94 | −10.10 |
| DR | 99°39′ | 33°45′ | 3967.5 | 1956–2015 | 551.7 | −0.79 | 6.90 | −6.88 |
| XH | 99°59′ | 35°35′ | 3323.2 | 1960–2015 | 362.8 | 1.44 | 9.80 | −5.79 |
| GL | 100°15′ | 34°28′ | 3719 | 1991–2015 | 512.4 | 2.25 | 9.07 | −6.76 |
| TD | 100°39′ | 35°16′ | 3289.4 | 1954–1998 | 424.2 | 0.51 | 10.00 | −7.13 |
| GN | 100°45′ | 35°35′ | 3120 | 1999–2015 | 439.9 | 2.89 | 11.76 | −3.81 |
| ZK | 101°28′ | 35°02′ | 3662.8 | 1957–1990 | 476.0 | −2.18 | 6.25 | −9.15 |
| HN | 101°36′ | 34°44′ | 3500 | 1960–2015 | 581.7 | 0.31 | 8.98 | −6.58 |
| JZ | 101°29′ | 33°26′ | 3628.5 | 1959–2015 | 748.1 | 0.74 | 9.41 | −5.71 |
| TR | 102°01′ | 35°31′ | 2491.5 | 1991–2015 | 403.5 | 6.52 | 14.0 | −1.01 |
| MQ | 102°05′ | 34°00′ | 3471.4 | 1967–2015 | 600.4 | 1.71 | 9.13 | −4.24 |
| LMS | 102°38′ | 34°05′ | 3362.7 | 1957–1988 | 779.9 | 1.21 | 9.00 | −4.49 |
| HZ | 102°54′ | 35°00′ | 2910 | 1957–2015 | 546.2 | 2.55 | 11.08 | −3.57 |

Note: ZXZ=Zhongxinzhan; DR=Dari; XH=Xinghai; GL=Guoluo; TD=Tongde; GN=Guinan; ZK=Zeku; HN=Henan; JZ=Jiuzhi; TR=Tongren; MQ=Maqu; LMS=Langmusi; HZ=Hezuo

**Table 2. Statistics of cross-validation and split-period calibration/verification**

| Calibration Period | r | $R^2_{adj}$ | F | p | Verification Period | ST/FST | PMT | RE | CE |
|--------------------|-----|-------------|--------|-------|---------------------|----------|-------|-------|--------|
| 1960–1991 | 0.470 | 0.194 | 8.228 | 0.008 | 1992–2014 | 22+**/18+** | 3.651 | 0.743 | 0.017 |
| 1984–2014 | 0.740 | 0.532 | 35.166 | 0.000 | 1960–1983 | 22+**/17+* | 2.945 | 0.677 | –0.589 |
| 1960–2014 | 0.767 | 0.580 | 74.187 | 0.000 | 1960–2014 | 45+**/38+** | 5.130 | 0.557 | 0.557 |

r = Pearson correlation coefficient; $R^2_{adj}$ = variance after adjusting; ST/FST = sign test/first difference sign test; PMT = product-mean test; RE = reduction error; CE = coefficient of efficiency.

**p < 0.01; *p < 0.05


