# Peer review of "The response of annual minimum temperature on the eastern central Tibetan Plateau to large volcanic eruptions over the period 1380–2014 AD"

_Climate of the Past, 2020_

## Referee Comment (RC1) · Anonymous Referee #1 · 14 Aug 2020

Line 34/35: Related to which events? Not really clear how minimums are related to 8,15, an 18 months. What causes interval cooling? A mechanism must have been discussed.

Line 55: Are these new samples or samples from the studies referenced in the sentence before? Were they published for another study? This must be clarified.

Line 55: Explain what a "strong" volcanic event is.

Line 61: The site name has "farm" in it which suggests that the area has had human activity and disturbance. Please elaborate.

[Figure]

Line 61: Give coordinates for the site.

Map: Need to zoom closer to the sites, currently some of the names are unreadable.

Line 63: Briefly describe rainfall amount and temperature. From the closest met station in fine. I see this information is in Table 1- this should be mentioned when you first discuss met data and seasonality Ĭline 63.

Line 81: Reference your map when talking about station locations.

Line 116: Some description of what SEA is would be helpful for readers.

Are there any longer met records to test? Even if they are further away? These met records are very short, with a lot of overlap between calibration/verification periods. Some of the met records here do not also show up in the US, NOAAs, Global Historical Climatology Network- why is that?

I would like to see a variant of the reconstruction using longer station records, even if they are father away, using just current JJ.

Line 131: Explain the obvious differences. It's better to explain it then to just say it is obvious.

Line 132: Refer to this as the 1st differenced chronology otherwise it is confusing.

Line 135: More information is needed in the introductory section about the growing season to understand if these correlations make sense.

Line 137: What do you mean by annual? P8-C7? That needs to be stated.

Figure 3. So much of the relationship between the trees and meteorological data is based on trend, as apparent by Figure 3b. Why not just reconstruct CJJ- the relationship (Fig 4b) must improve or look more convincing?

Section 3.5: This very large area of correlation is based on the warming trend across most of Asia, this needs to be stated more clearly. It is fine to show both, but Fig. C and

[Figure]

D are more representative of your results and reconstructed area, so I would suggest leading with that.

Line 185, Figure 6. The region that each Tmin timeseries is from needs to be labeled and mentioned in the text.

Line 188: The ratio description is unclear.

Figure captions could be more detailed.

Watch your tense- sometimes the text is written in past tense and sometimes in present tense.

---

## Referee Comment (RC2) · Anonymous Referee #2 · 11 Sep 2020

General comments: This study reconstructed the annual mean minimum temperature from the previous August to the current July using tree-ring width data collected at one site located in the eastern central Tibetan Plateau and analyzed the cooling effect of volcanic eruptions on the regional annual mean Tmin. The results of this paper is very interesting and important for further understanding of the dynamic mechanism of temperature variability in the study area. I recommend to publish this paper in Climate of the Past after a minor revision.

specific comments: 1. In this study, the extreme years was defined using the threshold of the mean plus/minus 0.5 SD of the reconstruction series. Under this criterion, it

seems that there are too many years are identified as extreme events. I hope the authors can give a consideration that whether it is plausible if the extreme years was defined based on the threshold of the mean plus/minus 1 SD/1.5 SD?

2. The statistics of calibration and verification show a relatively good skill for the regression model. However, Rbar values of the tree-ring width indices show a relatively large fluctuation over the whole period of 1380-2014. It is interesting that the highest Rbar values appear in two special intervals (i.e., the cold interval of 1800-1820 and the warmest period of 1980-2014). The cold period of 1800-1820 was driven by both the weak solar activity (Dalton Minimum) and a few very strong tropical eruptions (e.g., the Tambora eruption in 1815 and the other in 1809), while the warmest interval of 1980-2014 is likely a result of human activity. The cold period of 1800-1820 and the warmest interval since the 1980s were also found in the previous studies (both tree-ring-based temperature reconstructions and instrumental temperature data) on the Tibetan Plateau. I hope the authors can give a discussion on these two special periods based on the previous references.

3. The sentences in the paragraph 245 are need to be improved.

---

## Author Comment (AC1) · 22 Oct 2020

Please see the reply on comments with figures in supplement, the maintext are as below:

Anonymous Referee 1 Line 34/35: Related to which events? Not really clear how minimums are related to 8,15, an 18 months. What causes interval cooling? A mechanism must have been discussed.

Reply: Thank you for the comment. The eruption events indicated some large volcanic eruptions with dust veil indices >1000 m3. The citation here is an old article

that focuses on the relationship between temperature and volcanic eruptions but does not provide any information regarding mechanisms. We modified this sentences and added a new citation in the revision . Please see lines 33-41, page 2: "As early as 1985, studies of the relationship between large volcanic eruptions (dust veil index > 1000m3) and temperature variations from 1951 to 1980 in China (Zhang and Zhang 1985) demonstrated that temperature on the northeastern TP decreased 8, 15, and 18 months after eruptions. However, temperatures in the 6 months immediately following an eruption were found to be relatively high. Jia and Shi (2001) studied climate signals following volcanic eruptions from 1950 to 1997 and found that temperature on the TP dropped within 2 years after eruptions during this period. However, new research focusing on regional differences over China of winter temperature response to large volcanic eruptions with different latitudes and seasons from 1956 to 2005 found that winter volcanic eruptions led to extensive warming of winter temperature over Tibet Plateau (Sun et al., 2019)."

Line 55: Are these new samples or samples from the studies referenced in the sentence before? Were they published for another study? This must be clarified.

Reply: Thank you for the reminder. These are new samples taken from a different site in the Animaqin Mountains. Both the chronology data and the reconstruction series are used for the first time in this study. We have modified the sentence to clarify this; please see lines 59-60,Page 2: "Using tree-ring samples of Qilian juniper (Sabina przewalskii Kom.) collected from a new sample site in southeast part of the Animaqin Mountains, a 667-year tree-ring width chronology is developed, and then is used to reconstruct annual mean minimum temperatures (Tmin) across the east-central TP."

Line 55: Explain what a "strong" volcanic event is.

Reply: We have provided further clarification; please see lines62-63, Page 3: "Finally, this study explores the response of Tmin to strong volcanic eruptions (Volcanic Explosivity Index (VEI) $\geq$ 5) over the past six centuries."

[Figure]

Line 61: The site name has "farm" in it which suggests that the area has had human activity and disturbance. Please elaborate. Line 61: Give coordinates for the site. Reply: Thank you for pointing this out. There is no farm or history of human disturbance at this site. We have modified the site name to avoid further confusion and added the site coordinates and elevation.please see line 66-67, Page3. "Qilian juniper samples were taken from a natural woodland 35km west of Ningmute town, Henan County, Qinghai Province, China (E100.96, N34.62, 3806 m a.s.l)."

Map: Need to zoom closer to the sites, currently some of the names are unreadable.

Reply: Thank you for pointing this out. We have increased the font size in figure 1, as shown below:

Line 63: Briefly describe rainfall amount and temperature. From the closest met station in fine. I see this information is in Table 1- this should be mentioned when you first discuss met data and seasonalityâËŸA ËĞ Tline 63.

Reply: We have added a brief description in line 68-70, page3: "According to climate data of the Henan meteorological station (1960–2015; Table 1), mean annual temperature is 0.31°C and mean annual precipitation is 582 mm. Precipitation is mainly concentrated between May and September"

Line 81: Reference your map when talking about station locations.

Reply: We reference the map (figure 1) in the subsequent paragraph.

Line 116: Some description of what SEA is would be helpful for readers.

Reply: A brief discussion of SEA has been added in lines 124-132, page 5: "The Superposed Epoch Analysis (SEA) method (Haurwitz and Brier, 1981) was also used to analyze the influence of volcanic activity on regional temperature. SEA is a statistical method used to resolve significant signal-to-noise problems and is often used to study the link between climate and discrete events such as solar activity, fire events, volcanic activity etc. (Brad et al., 2003; Singh and Badruddin, 2006; Swetnam 1993; Esper et

al., 2013;). In this study, the year of a volcanic eruption is regarded as year 0. The years before the volcanic eruption are denoted as -1, -2, -3 and so forth, whereas the years after the eruption regarded as 1, 2, 3, etc.. The impact of volcanic eruption on temperature in the east-central TP was analyzed by comparing differences in temperature in the years leading up to and following an eruption."

Are there any longer met records to test? Even if they are further away? These met records are very short, with a lot of overlap between calibration/verification periods. Some of the met records here do not also show up in the US, NOAAs, Global Historical Climatology Network- why is that? I would like to see a variant of the reconstruction using longer station records, even if they are father away, using just current JJ.

Reply: We will address the two comments above together: Most instrumental records for the Tibetan Plateau start after 1956 because many weather stations were built later in this remote and cold place. The longest instrumental records available are from Yushu station (starting in October 1951 with many missing data until 1954), which is located more than 500 kilometers to the southwest of this site. The correlation coefficient of our chronology index with Tmin in current June-July at Yushu station is 0.62; the first-order differenced correlation coefficient is 0.39. These values indicate that the regression model has poor skill and does not explain much of the variation. More importantly, tree growth is directly affected by local environmental factors; at a distance of more than 500 km from the study site, data from Yushu station cannot be considered representative of the site conditions. In addition, this study focuses on temperature variations in the Animaqin Mountains and their response to volcanic eruptions. Thus, data from more distant stations are not suitable. There are three types of weather stations in China: national standard climate stations, national basic weather stations, and national general weather stations. Each type of station provides data of varying consistency and length. National standard climate stations are set up to obtain long-term and continuous climate data with sufficient representativeness. Records from these stations may be found in NOAA, Global Historical Climatology Network, such as

Dari station, Guinan station and Yushu station. The observation data obtained from the other two types of stations are mainly used by local and provincial meteorological services. These data are not reflected in the global climate observation system.

Line 131: Explain the obvious differences. It's better to explain it then to just say it is obvious.

Reply: We have modified this paragraph as follows: "The first-differenced correlations between the tree-ring width index and climate data are weak; in some months, the first-order correlations are even negative (Figure 3b). The first-differenced chronology correlates significantly and positively with precipitation of the previous September and the current May, but negatively with precipitation of the previous December."

Line 132: Refer to this as the 1st differenced chronology otherwise it is confusing.

Reply: Thank you, we have changed this as suggested.

Line 135: More information is needed in the introductory section about the growing season to understand if these correlations make sense.

Reply: We have added some additional information to the discussion section. Please see lines 213-224, Page 8. "With the thick topsoil and humid climate in spring and summer, the study area meets the needs of trees for radial growth. However, according to instrumental data from the GL weather station (Table 1), the elevation of which is close to that of the sampling site, the annual Tmean and Tmin are 2.25°C and -6.76°C, respectively. These temperatures are quite low for tree growth, and the statistically significant positive correlation between growth and temperature shows that tree radial growth in this area is restricted by temperature, especially Tmin. Tmin before and during the growing season may affect tree growth in several ways. In winter and early spring, warmer Tmin protects roots from cold damage and triggers earlier snowmelt. Warmer Tmin may therefore result in a longer growing season, and trees may experience increased growth in the subsequent growing season (Pederson et al., 2004; Fu

et al. 2012; Hollesen et al. 2015; Williams et al. 2015). In the other hand, Tmin is known to affect conifer tracheid division and enlargement by controlling the onset and conclusion of xylogenesis during the growing season (Deslauriers et al., 2003; Rossi et al., 2008; Li et al., 2017b; Hosoo et al. 2002; Steppe et al., 2015)."

Line 137: What do you mean by annual? P8-C7? That needs to be stated.

Reply: You are correct, annual Tmin means minimum temperature from previous August to current July. We have clarified this in the revision.

Figure 3. So much of the relationship between the trees and meteorological data is based on trend, as apparent by Figure 3b. Why not just reconstruct CJJ- the relationship(Fig 4b) must improve or look more convincing?

Reply: We attempted to develop the regression model using the Tmin of current June and current July, but the calibration/verification results were not good enough to warrant further use in reconstruction. We compared the instrumental and reconstructed Tmin in current June and July as shown below:(see in supplement)

Section 3.5: This very large area of correlation is based on the warming trend across most of Asia, this needs to be stated more clearly. It is fine to show both, but Fig. C and D are more representative of your results and reconstructed area, so I would suggest leading with that.

Reply: Thank you for the suggestion. We have modified this part and added a few sentences in the discussion section to clarifying the influence of warming trend on correlation results. Please see lines230-234, page 8: "The results of correlations of the reconstructed series with CRU Tmin reflect the regional significance of the reconstruction in general, however, the consistent warming trend of Tmin over most of Asia (Dong et al., 2017) may result in the large area of significant positive correlations, the results of first-order correlation analysis are therefore more referential, i.e. our reconstruction can reflect the temperature variations in the Animaqin Mountains and the area to the

west."

Line 185, Figure 6. The region that each Tmin timeseries is from needs to be labeled and mentioned in the text.

Reply: We apologize for the lack of clarity. Each Tmin time series belongs to one series. In the figure, we zoom in on different periods to more clearly present the relationship between volcanic eruptions and temperature variations in the series. We have modified the figure caption to make this clear.

Line 188: The ratio description is unclear.

Reply: We have modified the description for clarity. "A comparison of large volcanic eruption events and the reconstructed Tmin87 is shown in Figure 6. For most years, large volcanic eruptions (VEIsЁČ 4) coincide with a drop in Tmin across the study area. Of the 46 strong volcanic eruption years, there are 35 event year in which temperature decreases in the year of the eruption or 1, 2, or even 3 years after the event. "

Figure captions could be more detailed.

Reply: We have added more details to the figure captions.

Watch your tense- sometimes the text is written in past tense and sometimes in present tense.

Reply: Thank you; the tenses should now be consistent. When we refer to actions we took (e.g., we collected samples. . .), we use past tense. When we refer to the results (e.g., growth is positively correlated with. . .), we use present tense.

Please also note the supplement to this comment:
https://cp.copernicus.org/preprints/cp-2020-93/cp-2020-93-AC1-supplement.pdf

---

## Author Comment (AC2) · 22 Oct 2020

1. In this study, the extreme years was defined using the threshold of the mean plus/minus 0.5 SD of the reconstruction series. Under this criterion, it seems that there are too many years are identified as extreme events. I hope the authors can give a consideration that whether it is plausible if the extreme years was defined based on the threshold of the mean plus/minus 1 SD/1.5 SD?

Reply: Following your suggestion, we modified the threshold and the relevant sentences. Please see line 173-174 Page 6: "Tmin87 values below the mean 1.5ïĄşare defined here as 'extreme cold', and values above the mean 1.5ïĄş are 'extreme warm'.

There are 39 extreme cold years; the five coldest are 1488, 1490, 1824, 1862, and 1872. There are 47 extreme warm years; the five warmest are 1418, 1996, 1999, 2009, and 2010."

2. The statistics of calibration and verification show a relatively good skill for the regression model. However, Rbar values of the tree-ring width indices show a relatively large fluctuation over the whole period of 1380-2014. It is interesting that the highest Rbar values appear in two special intervals (i.e., the cold interval of 1800-1820 and the warmest period of 1980-2014). The cold period of 1800-1820 was driven by both the weak solar activity (Dalton Minimum) and a few very strong tropical eruptions (e.g., the Tambora eruption in 1815 and the other in 1809), while the warmest interval of 1980-2014 is likely a result of human activity. The cold period of 1800-1820 and the warmest interval since the 1980s were also found in the previous studies (both tree ring- based temperature reconstructions and instrumental temperature data) on the Tibetan Plateau. I hope the authors can give a discussion on these two special periods based on the previous references.

Reply: We have added a discussion about these two periods. Please see line 253-261, Page 9-10: "It is interesting that higher Rbar values appear in some special intervals, i.e., the cold intervals of the 1460s-1500s and 1800s-1820s and the warmest period of 1980-2014. These higher Rbar values indicate a good consistency among tree-ring series during these periods; in fact, these three intervals are evident in tree-ring-based studies from elsewhere on the Tibetan Plateau (Huang et al. 2019; Liang et al. 2016; Shi et al. 2019). The two cold periods identified in our series correspond to period of weaker solar activity (the Spörer Minimum and the Dalton Minimum), and to a few very strong tropical eruptions (e.g., the Tambora eruption in 1815 and another stratospheric eruption in 1809) (Cole-Dai et al., 2009). Similarly, the warming in 1980-2014 is closely related to the influence of human activities. These responses are indicative of the consistent response of tree growth to strong external forcing factors and of the reliability of our reconstruction."

[Figure]

3. The sentences in the paragraph 245 are need to be improved.

Reply: Thank you; we have modified this sentence in the revision. Please see line 268-273 Page 10. "On the southeastern TP, the cold period from 1816 to 1822 may have also been related to the Tambora eruption (Liang et al., 2008). Other research in the northeastern TP has shown that cold years can be matched with known tropical volcanic events in 15 of 21 cases (Zhang et al., 2014). We compared years of cooling we identified in this study with those identified by Zhang et al. (2014), the cooling years identified by the two studies are either the same or within a year of each other. On the southeastern TP, Liang et al. (2016) showed that the 15 coldest years of the past 304 years occurred mostly within 1-2 years of a major volcanic eruption, nine of these 15 cooling years are also seen in our study."

―――――――――――――――――――――――――

---

## Author Response (AR1)

In addition to the comments, questions, requests, and suggestions from the reviewers in the public review period, please respond to and make changes or additions as appropriate to the following items from the guest editor Kevin Anchukaitis:

Line 23: 'emit' instead of 'pour'
Line 45: 'studies' instead of 'researches'
Line 51: 'The'
**Reply:** *Thank you, we have changed all above as suggested.*

Line 102: VEI is not however always correlated with the climate-important sulfur emissions – what would happen if you used a list of eruptions based on the radiative forcing instead (e.g. from Sigl et al. 2015)?
**Reply:** *The global radiative forcing (GVF, Sigl et al. 2015) provides the GVF data in northern hemisphere (high-latitude), tropical and southern hemisphere (high-latitude) eruptions, and the GVF value varies greatly, thus, we analyzed the relationship between our Tmin reconstruction, all the data (i.e. all volcanic eruptions) and some lower data (GVF<-2.5, i.e. strong volcanic eruptions), in this case, there are not strong eruptions in southern hemisphere. the result is shown as follows:*

[Figure]

the SEA results of Tmin with GVF in global (a1: all data; a2: strong eruptions), northern hemisphere (b1: all data; b2: strong eruptions), tropical (c1: all data; c2: strong eruptions), and southern hemisphere(d)

*It seems that there is not significant relationship between the GVF series (volcanic eruptions) and temperature, except when the eruptions in southern hemisphere, temperature will be decreased in the next 3 and 5 years, however, after strong eruptions, Tmin will be decreased in the next 4 years. This result is different with ours, one reason maybe is that the timing and intensity of volcanic eruptions are different in two series. the VEI chronology from the Smithsonian institution includes more detailed information, such as longitude, latitude, starting and ending dates of volcanic eruptions et al., but the GVF series is not, for example, both the starting year (1812) and the ending year (1815) of the Tambora eruption(VEI 7) are shown in the VEI series, but the GVF series just provides a lower value of -17.2 in 1815, actually, the dates of other confirmed strong volcanic eruptions are different in two series, for examples, the Krakatau in 1883 (1883 in VEI series (VEI 6), 1884 in GVF series(GVF -17.2)), the Santa Maria in 1902 (1902 in VEI series (VEI 6), 1903 in GVF series (GVF -0.62)).*

Line 115: As also mentioned by Reviewer #1, more description is needed of the SEA procedure, including how the uncertainty/confidence levels were calculated. Also, are the SEA sensitive to the inclusion or exclusion of one or more events?

*Reply: We have added a further description in line 132-133, page5:*

*"The significance of responses is determined by a Monte Carlo resampling procedure (10,000 iterations) (Adams et al., 2003)."*

*In fact, The SEA method is one of the methods of mathematical statistics, which similar to calculating the average value of a series. The increase and decrease of sample size will lead to the change of degree of freedom, the larger the sample size is, the more stable the value is. however, Because the significance test is based on Monte Carlo method, after 10000 iterations based on Monte Carlo resampling procedure, the sensitivity of the results of significance test to the change of sample size should be not very large.*

Line 145: Preferable to report this either as R^2 of calibration or phrase it as 'The reconstruction account for 58% of the variance in the instrumental series during the calibration period'

*Reply: thank you for your comments, it has been modified as you suggested.*

Line 177: It is not clear what this sentence means - please rephrase for clarity or remove if necessary: 'If "high correlation" is defined by a correlation coefficient > 0.6, the high-correlation zone for the reconstructed Tmin87 is discontinuous and shrunken.'

*Reply: yes, we have removed this sentence in the revision.*

Line 187: I'm not sure what 'the number ratio' means? What are the units in the following line for 46/35 ? If this is simply the ratio of eruptions to years with a temperature decrease in Year 0 or Year 1, it would probably be better to express it the other ways (e.g. number of years with a temperature decrease in Year 0/1 vs. the number of VEI>5)

*Reply: yes, Reviewer #1 mentioned this problem too, we have modified this sentence in the revision, please see it as followed:*

*"For most years, large volcanic eruptions (VEIs> 4) coincide with a drop in $T_{min}$ across the study area. Of the 46 strong volcanic eruption years, there are 35 event year in which temperature decreases in the year of the eruption or 1, 2, or even 3 years after the event."*

Line 215: suggest rephrasing this to 'One reason for this could be a distortion of the meteorological ...' Alternatively this could be due to the detrending, could it not? You might mention here also that the low or negative value of CE should not influence your conclusions about the high frequency climate response to volcanism.

*Reply: thank you for your comments. Yes, the CE varies largely depending on the different chronologies with different detrending, however the current chronology is the best choice we can find. Actually, the different mean or trend in two split intervals could lead to the negative value of CE, the distortion of instrumental data may mainly lead to the lower value of CE, for example, the relocation of meteorological stations or the equipment replacement will result in the abrupt change of the trend or long-term mean in the instrumental data. We have modified these sentences according to the comments.*

Line 217: Suggest rephrasing this to: 'However, the cross-validation results indicate that the equation is otherwise reliable'

*Reply: it has been modified as you suggested. Please see it in line 229-232:*

*"One reason for this could be a distortion of the meteorological data due to the poor management and/or the relocation of meteorological stations during the 1950s and 1960s on the TP. However, the cross-validation results indicate that the equation is otherwise reliable, the negative value of CE should not influence our analysis about the high frequency climate response to volcanism.*
*"*

Line 227: Suggest rephrasing to: ' most strongly correlated '

*Reply: thank you, it has been modified as you suggested.*

Line 255: The reference from Cronin doesn't appear in the reference list but in any case this is not the best reference for this statement - the author might consider the recent paper by Toohey et al comparing the climate impact of extra-tropical and tropical eruptions, and/or the Schneider paper from 2009 which does idealized experiments with Community Climate System Model for tropical and extratropical eruptions. Indeed, I'm not sure this paragraph (through line 261) with a discussion of the stratosphere is really necessary to the discussion - better to ground this in modeling and observational studies of a more recent vintage

*Reply: thank you for your comments, we have modified this sentences as you suggested, please see it in line 279-288, Page 10:*

*"Studies have shown that some cold intervals in the eastern and southern TP may be influenced by large volcanic eruption in low-latitude regions (Bi et al., 2020; Duan et al., 2019a; Krusic et al., 2015), and the surface air temperature in the TP were cooling in the first winter based on the ensemble simulation of the climate response to high-latitude volcanic eruption (Oman et al., 2005). Using the fully coupled NCAR Community Climate System Model (CCSM3), Schneider et al. (2009) found there is significant cooling over the continents in summer in both scenarios (tropical and high-latitude volcanism cases) for the period 1250–1300 AD, and new research by Toohey et al. (2019) demonstrated extratropical explosive eruptions since 750 AD have produced stronger hemispheric cooling than tropical eruptions, these results are similar to ours, both strong eruptions in the northern hemisphere and tropical can lead to the cooling of Tmin in our study area, The effects of volcanic eruptions in the northern hemisphere seem to be more pronounced."*

Line 262: Suggest rephrasing to: 'influence local temperature'

*Reply: it has been modified as you suggested*

Line 273: Suggest omitting this sentence (it would be surprising if there were NOT interannual variability!): 'There is clear interannual variability in Tmin during the 1380–2014 AD period'

*Reply: thank you for your comment, it has been deleted as you suggested*

Figure 3: it would be preferable to use month names or abbreviations instead of numbers here if possible

*Reply: it has been modified as you suggested*

[Figure]

**Figure 3.** Correlations between (a) the tree-ring width index and climate data and (b) the first-differenced tree-ring width index and climate data. The horizontal solid lines indicate the 0.05 significance level; the horizontal dashed lines show the 0.01 significance level. Annu. indicates previous August to current July

Figure 6: I'm not convinced that this figure add to the argument in the paper - in particular, the arbitrariness of connecting eruptions to relative cold years 0 to several years after seems speculative and not well constrained statistically. More accurate would be to simply draw straight vertical lines associated with each volcanic eruption, and let the reader decide if the relationship in time seems plausible.

**Reply:** *it has been modified as you suggested*

[Figure]

**Figure 6. Plot of the VEI of large volcanic eruptions (black triangles) and cold years (black squares) since 1400 CE. The x-axis in each sub-figure indicates the year from the 15th century (top figure) to the 20th century (bottom figure). The dot lines and the numbers indicate the years of 46 volcanic events used in this study.**

Figure 7: Caption should also contain more information about how the significance levels were calculated here

**Reply:** *it has been modified as you suggested, the caption modified as follows:*

*Figure 7. SEA analysis of the impact of volcanic eruptions (VEI ≥ 5) on $T_{min}$ for the period 1380–2014 AD: (a) global; (b) Northern Hemisphere; (c) Southern Hemisphere; (d) low latitudes 30°S–30°N; (e) Northern Hemisphere mid-latitude 30°N–60°N ; (f) Southern Hemisphere mid-latitude 30°S–60°S; and (g) Northern Hemisphere high latitude. 0 = year of volcanic eruption; –1 = year before eruption; 1 = year after eruption. The solid lines represent 95% confidence limits using Monte Carlo type block bootstrapping (Adams et al., 2003).*

Dear Editor:

Thank you very much for your comments and considering our manuscript acceptance after minor revision.

According to your comments, we make the following modifications:

1. We slightly changed our conclusion in the abstract and conclusion, i.e. change "low latitude" to "northern hemisphere", in the revision, we find Tmin in our study significantly response to both strong eruptions in low latitude and NH, however, the influence of eruption in NH is more significant according to Fig 7, we are very sorry that it was the mistakes in the first version we did not find and modify it.

2. With regarding to editor comments, we made amendments one by one, please see them in the reply above or the marked-up revision below.

3. Because one paragraph in discussion section was rewritten according to the comments, we added some new references in the revision and deleted several references either.

4. According to the comments, we modified figure 3 and figure 6.

We have carefully addressed all of the editor's suggestions and concerns. Should further revision be needed, we are happy to work with you to further improve the manuscript. Thank you very much for your support and understanding.

Sincerely yours,

Yong Zhang, on behalf of all authors

[revised manuscript text omitted]